# Robust Indoor Localization Methods Using Random Forest-Based Filter against MAC Spoofing Attack

**DOI:** 10.3390/s20236756

**Published:** 2020-11-26

**Authors:** DongHyun Ko, Seok-Hwan Choi, Sungyong Ahn, Yoon-Ho Choi

**Affiliations:** School of Computer Science and Engineering, Pusan National University, Busan KS012, Korea; uyt1209@pusan.ac.kr (D.K.); daniailsh@pusan.ac.kr (S.-H.C.); sungyong.ahn@pusan.ac.kr (S.A.)

**Keywords:** indoor localization system, indoor localization, received signal strength, fingerprinting, MAC spoofing attack, random forest, convolutional neural network

## Abstract

With the development of wireless networks and mobile devices, interest on indoor localization systems (ILSs) has increased. In particular, Wi-Fi-based ILSs are widely used because of the good prediction accuracy without additional hardware. However, as the prediction accuracy decreases in environments with natural noise, some studies were conducted to remove it. So far, two representative methods, i.e., the filtering-based method and deep learning-based method, have shown a significant effect in removing natural noise. However, the prediction accuracy of these methods severely decreased under artificial noise caused by adversaries. In this paper, we introduce a new media access control (MAC) spoofing attack scenario injecting artificial noise, where the prediction accuracy of Wi-Fi-based indoor localization system significantly decreases. We also propose a new deep learning-based indoor localization method using random forest(RF)-filter to provide the good prediction accuracy under the new MAC spoofing attack scenario. From the experimental results, we show that the proposed indoor localization method provides much higher prediction accuracy than the previous methods in environments with artificial noise.

## 1. Introduction

As wireless networks and smart phones have become widespread, indoor localization systems (ILSs) started receiving much attention. While tracking the location of the user’s devices, indoor localization systems can provide the location information to the service client and the service providers in an indoor environment. For example, in an art gallery, indoor localization systems can provide the location of devices so that visitors can obtain a description of the artwork they are currently viewing. In addition, gallery operators can place artworks based on statistical information of the user’s location.

Especially, Wi-Fi is commonly used for indoor localization because the wireless access point (WAP) information can be used without additional hardware [1]. To predict the location of user, Wi-Fi-based indoor localization systems generally use the received strength signal(RSS) values of user’s device captured by multiple WAPs. Here, RSS is a measurement of the power present in a received radio signal. However, their usage is limited because Wi-Fi-based indoor localization systems show low performance in environments with natural noise such as shading and multiple path fading [2].

As a representative method to overcome the performance degradation of Wi-Fi-based indoor localization systems under the environment with natural noise, some studies proposed a method using channel state information (CSI) that contained more location-related information than RSS. Even though CSI can improve the localization accuracy of the Wi-Fi-based indoor localization systems, the usage in practical applications is limited because it requires the modification of the device [3,4].

To address such practical issue, studies on removing natural noise using traditional filters such as moving average filter and particle filter are proposed [5,6,7]. As shown in Figure 1A, after removing the natural noise using filters, the user’s location is estimated through a heuristic classification algorithm such as decision tree (DT) and random forest (RF). Moreover, most recent studies try to apply deep learning techniques to estimate the user’s location by learning the characteristics of the RSS values and natural noise as shown in Figure 1B [8,9,10]. Let us note that the performance of both methods significantly decrease under the environment with artificial noise caused by media access control (MAC) spoofing attack [11]. Here, MAC spoofing attack is an attack that attackers spoof their MAC address into MAC address of user’s device to perform a man-in-the-middle (MITM) attack.

In this paper, we introduce a specific MAC spoofing attack scenario, where the localization accuracy of the state-of-the-art indoor localization methods decreases. In this attack scenario, after spoofing the user’s MAC address, an adversary sends his signal to wireless access points (WAPs) distant from the actual location of the user as if his signal comes from a normal user’s device. As a result, RSS values with the same one as the user’s device ID are captured even at locations where the user device is not actually located. Such results can cause severe budget losses on discount stores which are sensitive to small rearrangement the interior.

To deal with this attack scenario using the artificial noise, we propose a new deep learning-based indoor localization method whose overall operation are shown in Figure 1C. Different from the previous deep learning-based methods in Figure 1B, the RF-based filter is applied to remove artificial noise before feeding into the deep learning model. The RF-based filter learns noise patterns of MAC spoofing attack. After identifying whether the RSS value includes artificial noise generated by MAC spoofing attack or not, the RF-based filter removes the artificial noises.

Main contributions of this paper can be summarized as follows: (1) After analyzing the problem of the previous indoor localization systems, we introduce a possible attack scenario decreasing their localization accuracy; (2) We propose a new deep learning-based indoor localization method using RF-based filter to show the good localization accuracy under the environment with artificial noise; (3) From the experimental results with multi-building, multi-floor dataset [12], we show that the deep learning-based indoor localization method shows better localization accuracy against MAC spoofing attack than the state-of-the-art deep learning-based indoor localization method.

Since we applied a Random Forest filter to remove artificial noise generated by MAC spoofing attack, this paper is similar to the work of Alotaibi et al [13]. However, in this work, Alotaibi et al. did not consider RSS time-series information. Different from the work of Alotaibi et al. by considering RSS time-series information, we consider when fake user’s signals are captured by the AP in a space different from the actual user during the time the indoor localization system estimates the user’s indoor location.

The rest of this paper is organized as follows. In Section 2, we introduce the existing works related to Wi-Fi-based indoor localization systems. In Section 3, we show some preliminary experimental results for designing the proposed indoor localization method. After describing the details of the proposed MAC spoofing attack scenario and deep learning-based indoor localization method in Section 4, we evaluate the performance of the proposed deep learning-based indoor localization method from the experimental results using a multi-building and multi-floor indoor localization dataset in Section 5. Finally, we conclude the paper in Section 6.

## 2. Related Work

According to the data collected from user’s mobile devices, the Wi-Fi-based indoor localization systems are categorized into using a CSI and using a raw RSS in general. In 2017, Hao Chen et al. proposed a convolutional neural network (CNN)-based Wi-Fi localization algorithm using a time-frequency matrix organized from CSI. After converting complex number information of CSI into a feature image, they showed about 91% accuracy through a CNN model consisting of three convolutional layers and two fully connected layers [3]. Shangqing Liu et al. used CNN to extract the relationship between the channel information of CSI and the number of people in the multi-human environment. Also, they used long short term memory (LSTM) model to analyze the dependence between the number of people and CSI. This method showed average accuracy by as much as 86.4% under the environment with five or more people [4].

However, Wi-Fi-based indoor localization systems using CSI have the limitation that the existing device driver should be modified. As an alternative, the Wi-Fi-indoor localization systems using raw RSS from WAP have been widely studied [14]. The indoor localization systems using raw RSS are mainly categorized into two groups: (1) filtering-based approach [5,6,7,15,16]; (2) deep learning-based approach [8,9].

The filtering-based methods remove noise before estimating the user’s location by using a classifier. Henri Nurminen et al. proposed running a light-weight fallback filter in the background of real-time particle filter and forward-backward recursions-based smoother for 2D, 3D door positioning [5]. Bodhibrata Mukhoopadhyay et al. suggested estimating a particular location using mode values of RSS and removing high frequency noise using moving average filter to improve indoor position accuracy [16]. Zhu Nan et al. proposed a new particle filter based on Rao Blackwellized particle filter (RBPF) to address the issue that WAPs could be sparse and short range [6]. In addition to improving performance using filters, the classification system for indoor positioning has been advanced through machine learning techniques. Rafał GÓRAK et al. proposed a modified random forest algorithm for indoor localization system [15]. They also showed that the indoor localization system worked without errors even in situations where some WAPs were turned off. Sunmin Lee et al. proposed a system that estimated the indoor location of smart watch devices using random forest, and used basic service set identifier (BSSID) as well as RSS to address the problem of similar signal strength [7]. The location-based radio map and BSSID list-based radio map is also used to reduce the number of comparison.

The deep learning-based methods train noise itself, such as shading and fading. Kim et al. proposed a hierarchical deep neural network (DNN) architecture consisting of a stacked autoencoder for multi-label classification of building, floor, and location [9]. By appending autoencoder models according to the number of buildings and floors, their DNN architecture for multi-building and multi-floor indoor localization can cover a large-scale complex of many buildings. To address the interference of moving objects and co-channel interference, Qiwu Zhu et al. proposed an ensemble model consisting of fuzzy classifier and multi layer perceptron(MLP) at the indoor parking localization [17]. They showed high accuracy through experiments using indoor parking lots in real shopping mall. Mai Ibrahim et al. presented a CNN-based method for indoor localization from multi-building and multi-floor dataset [8]. Their method showed 100% accuracy for building and floor prediction by using RSS time-series information. However, the performance of the deep learning-based methods decreases when fake RSS tuples artificially generated from active attacks such as MAC spoofing attack are injected into fingerprint DB.

## 3. Preliminary

In this section, we introduce the experimental environment used at Mai Ibrahim et al.’s method [8], which is a state-of-the-art CNN-based indoor localization using RSS time-series data, and shows the measured prediction accuracy using a public RSS dataset, called UjiIndoorLoc [12]. We also introduce the limitation of the deep learning-based method under a threat model through active attacks such as MAC spoofing attack. Such a limitation motivated us to design the proposed deep learning-based indoor localization method using RF filter.

In Figure 2, we show an example which shows the overall operation of how to predict user’s indoor location using RSS values in photo exhibition. Let us consider a user in front of the eagle photo whose position ID is one. First, the user’s mobile device sends the RSS values together with the phone ID(: 1). Then, nearby WAPs, i.e., WAP(A) and WAP(B), capture RSS. Second, WAPs (A) and (B) send the captured RSS and phone ID to a server(: 2). Third, after collecting RSS values from two WAPs and storing in the fingerprint DB, a server predicts position ID 1 matched with user’s location(: 3). Fourth, a server sends the narrative information related to position ID 1 to user’s mobile device(: 4, 5).

As described in [8], we generate a feature image using *T* number of RSS tuples with the same phone ID stored in the fingerprint DB for *S* seconds. For example, let us assume that *T* is set into 3 and RSS tuples are collected from 6 WAPs. As shown in Figure 3a, three raw time-series RSS tuples with phone ID 1 are used to generate a feature image, whose size is 3×6. Using the feature image, we train and test the deep learning-based method.

In practice, after implementing the experimental environment used at Mai Ibrahim et al.’s method [8], we measured the prediction accuracy of indoor localization using UjiIndoorLoc dataset [12], which has RSS values collected from 520 WAPs. From *S* and *T* values set into 60 and 3 respectively, we create the feature image whose size is 3×520×1, i.e., T×(# of WAPs)×1. Also, we train and test the deep learning-based method using 6453 number of training data and 138 number of test data, respectively. When training CNN model composed of two convolutional layers, we set the parameters into: ReLU, Softmax, Adam and MaxPooling for activation function, output layer activation function, optimizer, and pooling method, respectively [18,19]. Also, both the kernel size and the stride are set to 2. As a result of training with 30 epochs, the CNN model showed the indoor location prediction accuracy by as much as 94.93%. When *T* was set to 4, the accuracy were improved up to 100%.

Let us note that by targeting on the environment shown in Figure 2, as shown in Figure 3b, an adversary can generate a manipulated RSS tuple including artificial noise with the same phone ID. By injecting a fake RSS tuple(red-colored tuple) into fingerprint DB through active attack such as MAC spoofing attack, the adversary can cause the wrong prediction from a deep learning-based indoor localization method. That is, such manipulation results in poor indoor localization prediction accuracy. As shown in Figure 4, with the MAC spoofing attack, a user who requests some location information receives the wrong information from server. To prevent attacks from such a threat, refining RSS tuples stored in the fingerprint DB is necessary before generating the feature image. As an efficient method to eliminate artificial noise generated from such a threat, we propose a deep learning-based indoor localization method using RF filter.

## 4. Proposed Method

In this section, we describe the proposed deep learning-based indoor localization method using the RF filter to eliminate artificial noise. After introducing MAC spoofing attack scenario to generate a fake RSS tuple with the artificial noise, we show a novel defense method, which is a deep learning-based indoor localization method using RF filter.

### 4.1. MAC Spoofing Attack Scenario

As the user moves to the falcon photo from the eagle photo, the user’s mobile device sends RSSs to the surrounding WAPs (D), (E), and (F). What if the adversary sends the fake RSS data with the disguised user’s device ID to WAPs (A), (B), and (C)? As a result, the user receives information about the eagle, not the falcon, because the indoor localization system on the server returns an abnormal prediction result. To understand the targeted MAC spoofing attack scenario, let us consider an artificial noise injection example through MAC spoofing attack in Figure 5. First, after spoofing the MAC address of the targeted user’s mobile device at position 2, the adversary sniffs and analyzes RSS data from the user’s mobile device (: ①). Second, the adversary sends the fake RSS data with phone ID of the user’s mobile device (: ②). Third, WAPs (A), (B), and (C) nearby the adversary capture RSS. Third, WAPs (A), (B), and (C) send the captured RSS and phone ID to a server (: ③). Fourth, after collecting RSS values from three WAPs and storing in the fingerprint DB, a server predicts position ID 1 matched with the predicted user’s location (: ④). Fifth, a server sends the narrative information related to position ID 1, i.e., eagle information, to the user’s mobile device, not falcon information (: ⑤, ⑥).

Overall control flow for the artificial noise generation is shown in Figure 6. First, an adversary sorts the collected raw RSS data in ascending order by timestamp, building ID, floor ID, and phone ID. Second, the adversary generates each group data which has the same phone ID, building ID, Floor ID from the sorted data. Third, the adversary sets the start time, the end time and the targeted phone ID. Here, *S* is set into the end time minus the start time. Finally, the adversary may generate from 0 to 50 the fake RSS data with the targeted phone ID, the different building ID and floor ID. This is because the artificial noise can also be generated from multiple locations within the specific range of signal. As a result, as shown in Figure 6, the fake data is injected into fingerprint DB with the original group data. Therefore, the prediction accuracy of Wi-Fi-based indoor localization systems decreases severely.

### 4.2. Deep Learning-Based Indoor Localization Method Using RF Filter

Due to the existence of the fake RSS data, it is ineffective to use data in fingerprint DB for the indoor localization without refining. Indeed, these fake data make the highly performing indoor localization system less accurate. To address such an issue, we propose a new deep learning-based localization method, which applies the RF filter to remove the artificial noise from the RSS data.

In Figure 7, we show the overall operational procedure of the proposed indoor localization method with RF filtering. In contrast to the deep learning-based methods such as Mai Ibrahim et al.’s method where the original fingerprint DB is used, the modified fingerprint DB where the artificial noise is removed using RF filter is used. When the RSS tuples in the original fingerprint DB are given as the input data of RF, the independent *n* number of trees are used to classify the given input data. By conducting majority vote for the classified results from every trees, the final class, i.e., ‘fake’ or not ‘fake’, is made. If the final class for a RSS tuple is given into ‘fake’, the RSS tuple is removed and otherwise, the RSS tuple is kept.

## 5. Evaluation Result

In this section, we show how much the accuracy of the previous deep learning-based indoor localization method decreases under the introduced MAC spoofing attack scenario. Then, we show the performance evaluation result of the proposed deep learning-based indoor localization method for the artificial noise data. We also show the comparison results with the state-of-the-art indoor localization methods, such as noise training method and moving average filtering-based method, under artificial noise injected dataset.

### 5.1. Experimental Environment

To measure the performance of deep-learning-based indoor localization methods, the GPU server is used. The GPU server consists of an Intel(R) Xeon(R) CPU E5-2630v3 @2.40GHz with 8 cores, 62 GB RAM and an NVIDIA(R) GeForce RTX 2080 Ti. UjiIndoorLoc dataset [12] is used as the input dataset to evaluate the performance of the indoor localization method in multi-building and multi-floor environment. It contains RSS data collected from 520 WAPs, which are connected with 25 Android devices in the three buildings each of which has four floors. The RSS data has a negative integer values between –104 and 0 while +100 means that signal is not detected by a specific WAP.

### 5.2. Influence of Artificial Noise Injected Data on Indoor Localization System without Countermeasure

To observe the influence of the introduced MAC spoofing attack scenario on indoor localization methods without refining the artificial noise, we measured the accuracy of Mai Ibrahim et al.’s method described in Section 3. Before generating the feature images from the fingerprint DB, we replaced the RSS value of +100 into −110 and then, performed normalization before feeding it into CNN model.

In Figure 8a,b, we show the validation accuracy of CNN model for the original validation dataset and the artificial noise injected dataset, respectively. While the validation results using the original dataset show almost the prediction accuracy by as much as 94.93% on average, the validation results using artificial noise added data show the prediction accuracy by as much as 2.27% on average.

### 5.3. Indoor Localization Accuracy of the Proposed Method

To evaluate the proposed deep learning-based indoor localization method using RF filter, we measured the prediction accuracy of the following three indoor localization methods: (1) proposed deep learning-based method with RF filter; (2) deep learning method using artificial noise training [20]; and (3) filtering-based method using moving average filter [21].

#### 5.3.1. Accuracy of the Proposed Indoor Localization Method with RF Filter

To evaluate the performance of the proposed indoor localization method, we trained the RF model to find the artificial noise injected data through MAC spoofing attack. From the experiments where RF consists of 100 estimators (decision trees), the proposed indoor localization method using RF filter showed the indoor localization prediction accuracy by as much as 95.31% at maximum as shown in Table 1 and by as much as 94.81% on average as shown in Figure 9a. This result implies that the proposed indoor localization method using RF filter can successfully remove the artificial noise with high accuracy.

In Table 2, we also measured the performance of the proposed indoor localization method under various signal-to-noise ratio (SNR) situations where there is more fake data added by noise injection attacks. In this situation, the performance of the indoor localization system can be further worsen since the number of fake data is much more than the number of original real data. For high SNR situation where from 0 to 50 fake data was generated, the proposed indoor localization method using RF filter showed the indoor localization prediction accuracy by as much as 95.31% at maximum. For low SNR situation where from 50 to 100 fake data was generated, the proposed indoor localization method using RF filter showed the indoor localization prediction accuracy by as much as 94.68% at maximum. These results imply that the proposed indoor localization method using RF filter is effective in both high SNR and low SNR environments.

#### 5.3.2. Accuracy of Deep Learning-Based Method Using the Artificial Noise Training

To measure the performance of the deep learning-based method using artificial noise training, we train a CNN model using the artificial noise injected dataset instead of the original dataset. Similar to adversarial training [22], after adding artificial noise to the original RSS data, we train the CNN model. From Table 1 and Figure 9b, we observe that the deep learning-based method training with artificial noise injected dataset provides the prediction accuracy by as much as 54.46% at maximum and by as much as 53.77% on average, which are way lower than the proposed method, under artificial noise injected dataset. This is because even though the deep learning-based method was trained using the artificial noise injected dataset, the prediction accuracy decreased because RSS data appeared in multiple locations at the same time. For example, if the user is in location *A*, only WAPs around location *A* should have a RSS value between −104 and 0 while others should not. However, if WAPs around another location *B* have a value between −104 and 0 due to the artificial injected noise, the CNN model will be confused due to such conflicting RSS values collected from WAPs located in different locations at the same time. This result implies that while the deep learning-based indoor localization method using the artificial noise training can comprehend natural noise such as shading and fading, the artificial noise generated from MAC spoofing attack cannot be identified.

#### 5.3.3. Accuracy of Filtering Method Using the Moving Average Filter

As one of the most common signal filters in digital signal processing(DSP), the moving average filter has been often used in many practical applications. For successive data, the moving average filter estimates the average value for the given window size. After setting the window size to 5, a dataset is generated with the average value of each five consecutive RSS data and fed into a CNN model. As shown in Table 1 and Figure 9c, the filtering-based method using moving average filter showed the lowest prediction accuracy by as much as 0.57% at maximum and by as much as 0.55% on average under artificial noise injected dataset. This result implies that likewise the deep learning-based method using the artificial noise training, the filtering-based method using moving average filter is effective when reducing natural noise, but is not effective when reducing artificial noise generated from MAC spoofing attack.

## 6. Conclusions

Indoor localization systems have been implemented using various technologies such as Wi-Fi, RFID, Bluetooth, and so on. Among them, Wi-Fi technology is commonly used for indoor localization systems because it does not need any additional hardware. The existing Wi-Fi-based indoor localization systems use filtering or deep learning techniques to remove natural noise such as shading and fading. However, the previous methods are vulnerable to artificial noise generated from active attacks such as MAC spoofing attack. In this paper, we introduced a MAC spoofing scenario which generates the artificial noise injected data. In our MAC spoofing scenario, an adversary sends his signal to WAPs distant from the actual location of the user as if his signal comes from user’s device. The evaluation results show the prediction accuracy of the previous Wi-Fi-based indoor localization system without filtering decreased from 94.93% to 2.27%. To address the performance degradation problems due to the artificial noise, we also proposed a new deep learning-based indoor localization method using RF filter. The RF filter in proposed deep learning-based indoor localization method learns noise patterns of MAC spoofing attack to identify and remove artificial noises. From the experimental results using a public dataset, we showed that the proposed indoor localization method increased the prediction accuracy from 2.27% to 95.31% under the artificial noise injection attack.

## Figures and Tables

**Figure 1 sensors-20-06756-f001:**
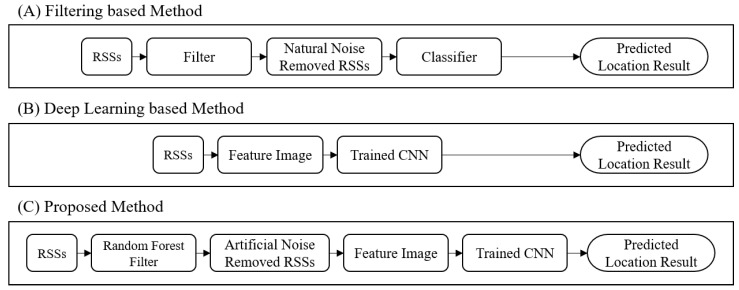
Operational procedure of various indoor localization methods: (**A**) Filtering-based method; (**B**) Deep learning-based method; (**C**) Proposed method.

**Figure 2 sensors-20-06756-f002:**
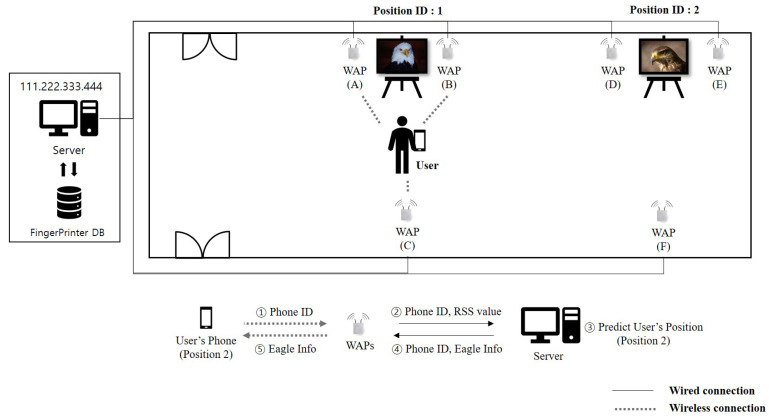
An example of using RSS values to predict user’s indoor location in photo exhibition.

**Figure 3 sensors-20-06756-f003:**
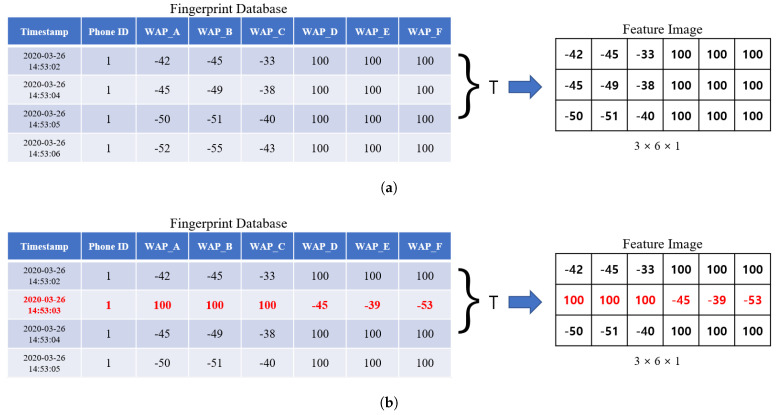
Examples of RSS values in fingerprint DB and the corresponding feature map, where *T* = 3, the number of WAPs is 6, and +100 in DB means that that signal is not detected by a specific WAP. (**a**) For only raw time-series normal RSS values. (**b**) For raw time-series normal and artificial(red-colored) RSS values generated through MAC spoofing attack.

**Figure 4 sensors-20-06756-f004:**
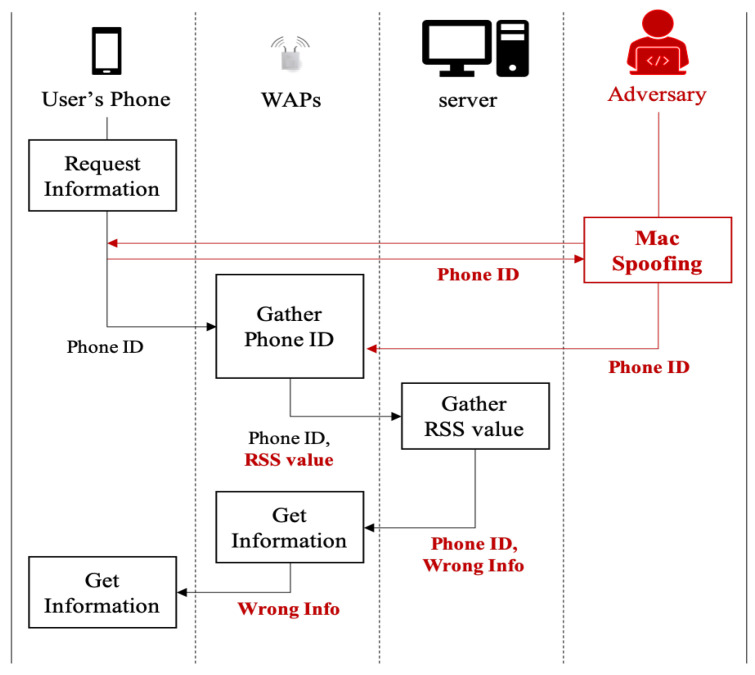
Overall flowchart of a MAC spoofing attack scenario on indoor localization system.

**Figure 5 sensors-20-06756-f005:**
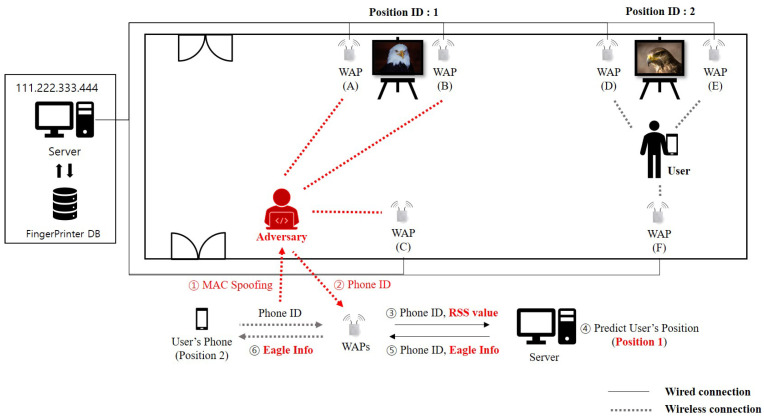
An artificial noise injection scenario through MAC spoofing attack.

**Figure 6 sensors-20-06756-f006:**
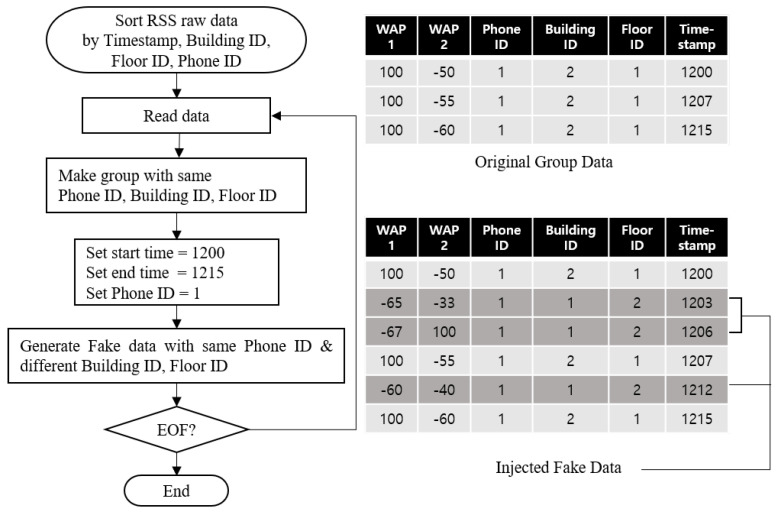
Overall control flow of artificial noise generation method with an example.

**Figure 7 sensors-20-06756-f007:**
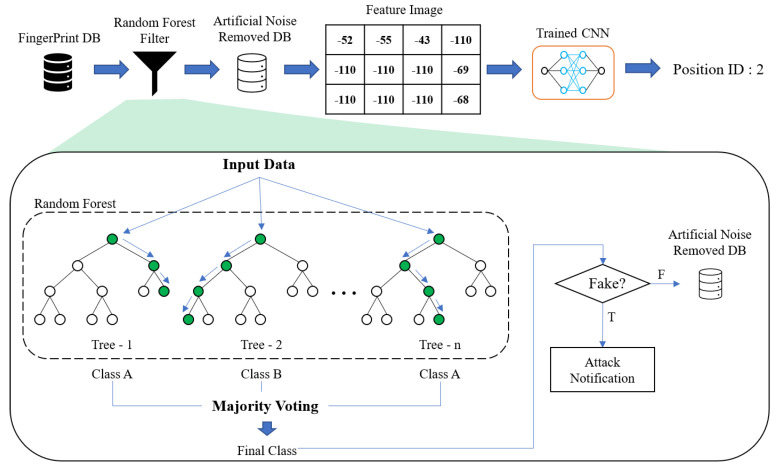
Overall operation of the proposed indoor localization method using RF-based filter.

**Figure 8 sensors-20-06756-f008:**
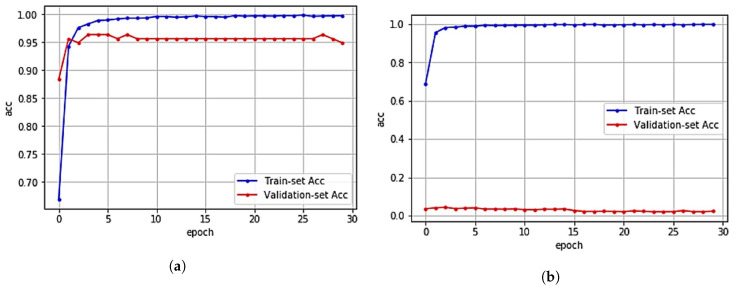
Accuracy of CNN model under (**a**) the original data and (**b**) the artificial noise injected data. (**a**) Validation result with the original data. (**b**) Validation result with the artificial noise injected data.

**Figure 9 sensors-20-06756-f009:**
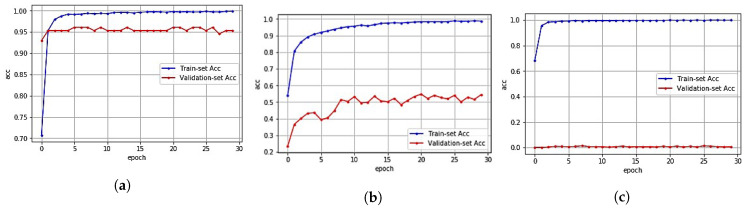
Validation results of three indoor localization methods for artificial noise injected data under various epochs. (**a**) Proposed method with RF filter. (**b**) Deep learning-based method using rtificial noise training [20]. (**c**) Filtering-based method using moving average filter [21].

**Table 1 sensors-20-06756-t001:** Maximum accuracy comparison of three indoor localization methods for artificial noise injected data.

	Proposed Method	Deep Learning-Based Method	Filtering-Based Method
	with RF Filter	Using Artificial Noise Training [20]	Using Moving Average Filter [21]
Max. Validation	95.31%	54.46%	0.57%
Accuracy			

**Table 2 sensors-20-06756-t002:** Maximum accuracy comparison of proposed method on noisy environment.

	Proposed Method with RF Filter	Proposed Method with RF Filter
	in High SNR Environment	in Low SNR Environment
Max. Validation	95.31%	94.68%
Accuracy

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
