# Peer review of "Robust Indoor Localization Methods Using Random Forest-Based Filter against MAC Spoofing Attack"

_sensors, 2020, doi:10.3390/s20236756_

Round 1

Reviewer 1 Report

The paper focuses on indoor localization, which is a hot topic. Overall the paper is well written while the proposed method is reasonable. My comments are listed below:

  1. The comparison with existing methods is missing, experiments on state-of-the-art methods is required to fully evaluate the proposed method.
  2. The language can be improved, I would not go against the idea to carefully read it several times.
  3. Some related work are missing, e.g., a. RoArray: Towards more robust indoor localizaiton using sparse recovery with commodity WiFi, IEEE Transactions on Mobile Computing 2019; b. Multiview and multimodal pervasive indoor localizaiton, ACM multimedia 2017;  c. Continuous space estimation: increasing WiFi-based indoor localization resolution without increasing the site-survey effort, Sensors 2017.

Reviewer 2 Report

1- The author presents an Robust Indoor Localization Methods Using Random Forest-based Filter against MAC Spoofing Attack as the author has "copied" a large portion of this manuscript from "A New MAC Address Spoofing Detection Technique Based on Random Forests" without mention it in there references 2- the problem statement should be more clear and at least they should add algorithm steps or flowchart 3- figures ,7, 8 should be more clear 4- the conclusion part looks like more an abstract rather than a conclusion

Reviewer 3 Report

The paper presents an interesting method to deal against MAC spoofing attack.

In general, the paper is very well written and the method is well explained. The results demonstrate the good performance of the method.

I have just three minor issues:

1) I suggest adding some text in the introduction explained in which situations a MAC spoofing attack can be produced. It is not easy for me to understand why a person would like to perform this kind of attack in the art gallery scenario. 

2) The quality of the figures (e.g. Figures 7 and 8) must be improved. 

3) I suggest testing the method in some other scenarios in addition to UjiIndoorLoc, for instance, the Tampere University one or even some another more recent.

Round 2

Reviewer 2 Report

I don't have any more concerns